# The Oral Health Status of Spanish Naval Military Personnel: A Retrospective Study

**DOI:** 10.3390/jcm14155236

**Published:** 2025-07-24

**Authors:** Bárbara Manso de Gustín, Alfonso Alvarado-Lorenzo, Juan Manuel Aragoneses, Manuel Fernández-Domínguez

**Affiliations:** 1Department of Translational Medicine, CEU San Pablo University, Urbanización Montepríncipe, 28925 Madrid, Spain; jmaragoneses@gmail.com (J.M.A.); clinferfun@yahoo.es (M.F.-D.); 2Capt. (OF-2) Spanish Navy, Enfermería B.N. Rota (Clínica Dental), Carretera de Rota s/n, 11530 Cádiz, Spain; 3Department of Surgery, Faculty of Medicine, University of Salamanca, 37008 Salamanca, Spain; alfonsoalvaradolorenzo@gmail.com; 4Department of Dental Research, Federico Henriquez y Carvajal University, Santo Domingo 10106, Dominican Republic; 5Facultad de Odontología, Universidad Camilo José Cela, C/Castillo de Alarcón, 49 Urb. Villafranca del Castillo, Villanueva de la Cañada, 28691 Madrid, Spain

**Keywords:** oral health, dental caries, DMFT, dental care, care index, navy, military personnel

## Abstract

**Background/Objectives**: Oral health has specific importance and consequences from a military and Navy standpoint. The aim of this study was to determine and compare caries prevalence and dental care in Spanish Navy personnel. **Methods**: A retrospective observational study was carried out with a sample size of 1318 individuals (34.65 ± 8.82 years old) stationed at the Rota naval base in Spain, whose dental charts were examined. Caries prevalence was assessed using the Decayed, Missing, Filled Teeth (DMFT) index; dental care was evaluated using the care index (CI); and demographic and occupational factors were recorded. **Results**: The population of this study had a mean DMFT index of 5.99 ± 4.71 and a CI of 79%. Through the results of the DMFT index and CI, the statistical significance of the age and rank variables (*p* < 0.01) was confirmed, with personnel >45 years old and non-commissioned officers (NCOs) having the highest mean DMFT values and the youngest and officer groups having the greatest CI variable. Comparing the DMFT index across genders and ages and between age and rank also revealed significant differences. **Conclusions**: This study’s findings show a low prevalence of cavitated caries (14.5%), with intermediate DMFT values and higher CIs compared to those in previous published data.

## 1. Introduction

The repercussions of oral diseases on quality of life and general health constitute significant components of overall disease burden. Oral diseases serve as clinical indicators of suboptimal oral health and include tooth decay and tooth loss, the latter being one of the most frequent consequences of periodontal disease [1,2,3].

Dental caries is a multifactorial disease that affects the tissues of teeth, primarily caused by the interaction of microorganisms present in dental plaque over time. It continues to be one of the most prevalent chronic diseases in adults [4,5], affecting a significant number of people around the world and being able to cause pain, tooth loss, and even the need for surgery [6,7]. Tooth loss is frequently incorporated into extensive epidemiological studies and utilized as a significant marker in oral health assessment [3].

On a global scale, the burden of dental caries remains alarmingly high. Despite advances in dental care and preventive measures, the prevalence of untreated caries in permanent teeth has remained virtually constant since the 1990s [1,8]. According to the 2015 Global Burden of Disease (GBD) study, untreated caries lesions in permanent teeth were the most prevalent condition, affecting approximately 30% of the global population [9,10]. In the 20th century, the management of caries lesions shifted to controlling them and halting the progression of early signs, rather than focusing on restorative care [11]. Nonetheless, the prevalence of adult dental caries is elevated worldwide, with almost 100% of the population affected in most countries [1,6,9,10,12]. In nations such as Spain, the prevalence of dental caries remains high, with an average Decayed, Missing, Filled Teeth (DMFT) index of 10.9 and a care index (CI) of 27%; particularly, in the 35–44 age group, the DMFT is 7.40 and the CI is 63.1% [13,14]. Dental caries pathologies are the primary causes of oral morbidity, both in Spain and internationally [15].

The World Health Organization (WHO) proposed a caries detection system, which is the most extensively used in epidemiological studies. It is aimed at assessing the prevalence of caries in diverse population groups and has two key objectives regarding oral health: reducing morbidity and reducing disparities. Caries experience is typically measured using the DMFT index, which calculates the total number of teeth that are decayed (D) or missing (M) due to caries and filled (F) teeth. To assess the prevalence and impact of dental caries in military populations, the DMFT index is commonly used, as it provides a comprehensive measure of caries experience [2,16,17]. The care index (CI), a supplementary measure, is expressed as a percentage of the total DMFT index and is a key indicator of the effectiveness of preventive and restorative dental care provided to individuals. Higher CI values reflect better care outcomes and a lower risk of morbidity [13,18]. These indices are essential for understanding both the extent of oral disease and the effectiveness of interventions within military populations.

Due to the circumstances they operate under, military personnel are considered a high-risk group for oral diseases, which have specific importance and consequences in this context. This is due to the direct and indirect repercussions affecting service resulting from a loss of oral health. One of the objectives of military training for soldiers and sailors is achieving the necessary fitness for missions to be carried out, and participation in maneuvers and deployments necessitates adequate oral health [19]. Military dentists are responsible for the oral health of military personnel. Their duties include conducting dental assessments, implementing preventive, legal, and forensic dentistry, and providing treatment when required. They are also responsible for deployment tours, either on land or on board naval vessels, under national, North Atlantic Treaty Organization (NATO), or EU mandates [20,21]. All military personnel are required to undergo a dental examination by a military dentist before deployments to ascertain their dental fitness category, which is a NATO-standardized risk assessment for oral health status whose objective is to minimize and reduce dental emergencies. Dental treatment is partly mandatory in terms of ensuring dental fitness, which is a prerequisite for combat aptness [2,20,22].

Within the military context, particularly in the Spanish Armed Forces, oral health holds paramount importance [15]. Military personnel are at a higher risk of oral health problems due to the physical demands of their service and limited access to dental care during long deployments. Specifically, naval personnel face unique challenges, such as prolonged deployments, intense work schedules, and restricted access to healthcare, all of which can have a detrimental effect on their oral health. Maintaining good oral health is essential for ensuring optimal performance during operations and missions, and dental health assessments are mandatory prior to deployment [20,21,22,23].

Despite the importance of oral health in military settings, there are limited investigations of oral health in military personnel; specifically, regarding the Spanish Armed Forces, there are almost no published epidemiological reviews. To our knowledge, there are no epidemiological studies that analyze caries experience in Spanish Navy professionals in the literature. Research on oral health within this population is needed for planning and establishing priorities for the management of preventable and treatable oral health conditions that could have a negative impact on their deployment.

The aim of this study is to evaluate oral health in Spanish Navy personnel and, more specifically, to analyze the prevalence of caries and measure restorative care received. The null hypothesis of this research is that there are no significant differences in terms of caries experience and restorative care among the different groups of the population studied.

## 2. Materials and Methods

### 2.1. Sample Size Calculation and Participants

In this study, a total of 1318 participants were included, with an age range between 21 and 56 years. Data were collected from the dental records of military personnel stationed at Rota Naval Station. To ascertain whether the sample size of this study was adequate, the GPower tool was used. The potential analysis value was determined to be 1.00 in all instances, thus indicating that the sample size was adequate for this type of study.

The inclusion criteria were as follows: active-duty military personnel belonging to the Navy and stationed at units of Rota Naval Station. The exclusion criteria were as follows: personnel from the Army or Air Force, reservists, retired personnel, civilian employees, and scuba divers.

This research project was approved by the Medical Research Ethics Committee (CEIm) of the HM Hospitals (2377-GHM) in Madrid, Spain.

### 2.2. Study Design

This was a retrospective study involving a representative random sample of Spanish Navy personnel, conducted between October 2020 and November 2023, to assess oral health status. The study population included military personnel assigned to the “Rota Naval Station” Navy base (Rota) and between 21 and 56 years of age, with ranks ranging from Seaman to Admiral.

A census sampling approach was employed, in which all available dental records of military personnel assigned to Rota Naval Station during the study period were included. While this did not encompass the entire population of military personnel at the station, the sampling method covered the majority of dental records available at the time. This allowed for the calculation of the Decayed, Missing, Filled Teeth (DMFT) index and care index (CI).

In this study, we utilized an anonymized dataset from the Rota naval base Dentistry Service’s patient healthcare records. Baseline data were extracted to obtain demographic information (age, gender, service, rank, place of birth, and unit).

Dental examinations were conducted by a single expert examiner, who is highly experienced in dental diagnostics. These procedures followed standardized clinical protocols to ensure consistency across the assessments, including those in the World Health Organization (WHO) guidelines for epidemiological surveys, such as the use of a WHO dental probe and mirror. Although intra-examiner reliability was not formally evaluated, the examiner’s extensive experience and the use of standardized procedures ensured a high level of reliability and consistency in the assessments.

The dental records used in this study were selected based on their availability during the study period. All records from military personnel stationed at Rota Naval Station who underwent mandatory pre-mission dental exams were included. This non-randomized approach provided a representative sample of the dental health status of personnel assigned to the base.

Caries history was evaluated using the DMFT index, which was calculated through the summing of ‘D’ (decayed teeth), ‘M’ (teeth missing due to decay), and ‘F’ (filled teeth), considering all available permanent teeth. Additionally, the CI was calculated by determining the number of restored teeth as a proportion of the total number of decayed (D), missing (M), and filled (F) teeth, using the following formula: CI = F/DMFT × 100%. The range of DMFT values is between 0 and 28 because wisdom teeth are not included in this index. The severity scale employed for the categorization of the mean DMFT values was based on the criteria established by the World Health Organization (WHO), which include the following: very low (<5.0), low (5.0–8.9), moderate (9.0–13.9), and high (>13.9). Evaluation was performed according to the WHO instructions for caries diagnosis in epidemiological surveys, using a dental mirror and a WHO dental probe for caries detection [16].

### 2.3. Statistical Analysis

The latest edition of the IBM SPSS Statistics program (version 29.0, Chicago, IL, USA) was utilized for the execution of statistical analyses.

Descriptive variables (mean and standard deviation) were obtained for all variables (CI and DMFT index) within each group. The means of each group of variables were compared using one-factor ANOVA, with *p*-values less than 0.01 (*p* < 0.01) being considered highly significant and those less than 0.05 (*p* < 0.05) being considered significant. Subsequently, post hoc comparisons were conducted for variables that demonstrated significant values. For the qualitative variables, cross-tables were created with their respective chi-squares. As previously indicated, a *p*-value of less than 0.01 was deemed to be highly significant, and that of less than 0.05 was considered to be significant.

## 3. Results

In the results of the DMFT variable, statistically significant differences are observed across several demographic and occupational categories. The statistical significance of the age variable (*p* < 0.001) is confirmed, with the >45 group having the highest mean (8.33 ± 5.14). Additionally, the place of origin variable is found to be statistically significant (*p* = 0.019), with those outside Europe having the highest mean (9.50 ± 5.32). For the rank variable, a significant difference is also confirmed (*p* ≤ 0.001), with non-commissioned officers having the highest mean (6.78 ± 4.82). The least significant difference (*p* = 0.046) exists for the unit variable, with other ships having the highest mean (7.30 ± 4.56). The effect sizes for all variables are small (0.01–0.10), indicating minimal practical significance, with age showing the largest effect (Table 1).

The differences between the groups according to rank, gender, and age in the DMFT variable were analyzed. When DMFT scores are compared between the gender and rank groups, a significant difference is found in males by rank (*p* < 0.001), with troops having the highest mean (5.88) (Table 2). Specifically, in the male group, there are differences between the values of different ranks (troops, non-commissioned officers, and officers) in all three combinations (Table 3). While the differences are statistically significant, the effect sizes indicate limited practical significance. Significant differences in DMFT scores are observed across age groups for both males (*p* = 0.004) and females (*p* < 0.001), with personnel >45 years having the highest scores (Table 4). Additionally, significant differences by rank are observed in the 20–34 and 35–44 age groups, with troops showing higher scores (Table 5).

In the results of the CI, several variables show significant differences between groups. The effect size for age is moderate, with the 20-to-34 age group having the highest mean (83.70 ± 26.47). The effect size for rank is also moderate, with officers having the highest mean (87.22 ± 24.91), confirming a notable difference in CI scores between ranks (Table 6).

Finally, the following tables analyze the differences in means between groups regarding different variables (age, place of origin, and rank), comparing them between the two measures, the DMFT index and CI. A substantial discrepancy in the means of the officers and those of the other two groups is evident, suggesting a significant difference according to rank. With regard to the place of origin, no significant differences are observed. Finally, with regard to age, there are significant differences in the means between the first age range (20–34) and the others (Table 7).

## 4. Discussion

This study analyzed and compared caries experience and dental care levels in different Spanish Navy populations. In the literature, there are some studies that have analyzed oral health in military personnel [2,24,25]. Fewer studies have focused on oral health in Spanish military staff [15,26,27,28]. Notably, no previous studies have specifically investigated the caries experience and care index in the Spanish Navy. In the literature, we found studies that analyze periodontal health in the Spanish Army [15,26] and the incidence of barodontalgias in military flight staff [28], as well as DMFT studies involving the Spanish Army in 1987 and 1994 [29,30]. Regarding caries experience, most of the studies found analyzed the DMFT index [2,23,24], whereas ones that study the CI are scarce [27]. Therefore, the purpose of this study was to compare the DMFT index and care index by evaluating demographic and occupational factors.

Untreated caries in permanent teeth have been reported to be the most prevalent medical condition worldwide [9,10]. This study found an unheralded and encouraging low prevalence of cavitated caries (14.5%), mainly when compared to the prevalences reported globally (29.4%), for employed adults in Spain (39.5%), for Spanish Army personnel (40,1%), and in another epidemiological study in Spain (31.9%) [10,14,27,31]. These results may reflect specific oral health considerations related to operational demands in the Navy and the emphasis on maintaining oral health status as a prerequisite for deployment. This indicates a higher level of readiness from a clinical and military perspective, reducing the risk of dental emergencies during missions [19,22].

The study population showed a low mean DMFT index of 5.99 ± 4.71, an intermediate value compared to previous published data [23,29,30]. This suggests moderate caries burden in the Spanish Navy, positioned between other national and military values. This figure is higher than the scores of 1.95 ± 2.67, 4.59 ± 4.24, 4.1 ± 4.2, and 3.0 ± 3.7 reported in the studies of the Israel Defense Forces by Yavnai et al. [32], Malaysian Armed Forces by Azis et al. [23], Finnish Defense Forces by Tanner et al. [33], and New Zealand Defense Forces by Naysmith et al. [24], respectively; however, it is lower than the score of 13.7 ± 0.4 reported in a study of the U.S. Armed Forces by Schindler et al. [34]. However, detailed international comparisons were minimized to focus on findings with direct relevance to our population. This study presents low DMFT scores compared to previous published data in the Spanish population; Martinez et al. [31] reported a higher DMFT index (8.2 ± 5.6) than that in our results, which was also similar to that reported in another epidemiological study in Spain (7.40 ± 4.86) [14]. Additionally, Spanish military personnel exhibited similar DMFT scores to those in our results in another study [27].

The dental care level of the study groups was assessed according to the mean care index values expressed as F/DMFT × 100. Clinically, the care index (CI) serves as a strong indicator of treatment coverage and reflects access to dental services and adherence to oral healthcare policies [18]. The population of this study had a mean CI of 79%, which was greater than those in previous published investigations of the Spanish population [14,27,35]. This high CI reflects a well-functioning military dental system and suggests a greater level of dental readiness. In military terms, such a level reduces the likelihood of untreated dental disease leading to operational disruption. An explanation for this greater care index score may be the availability of dental services and the exceptionally greater requirements for deployments.

Significant differences in age, rank, unit, and place of origin were observed across the DMFT index (Table 1) and in groups of age and rank across the care index (Table 2), likely reflecting underlying social determinants of health even within a structured system like the military and Navy. The variation in the DMFT and CI scores indicates that military and naval personnel encounter special oral health challenges, possibly because of their own considerations and lifestyle.

Most studies focus on cohorts of teenagers or young adults, due to a lack of epidemiological research on adults. In the present study, the mean (SD) DMFT index observed was 4.60 (4.10), 7.13 (4.64), and 8.33 (5.14) in subjects 20–34 years old, 35–44 years old, and older than 45 years, respectively (Table 1), with statistically significant differences among all three age groups (Table 3), highlighting the cumulative nature of dental caries over time. In contrast to a previous study on employed adults in Spain by Martinez et al. [31], our results show a lower DMFT index in each age group; however, in both studies, scores are higher as age increases. Nevertheless, the caries index for the 35–44 years group studied was similar to that in the last national survey [14]. Turning to the care index, statistically significant differences were found between the younger group and other age groups (Table 3). For the 35–44 years group, the CI was 75.69%, greater than values associated with Spain in investigations with a global perspective [13,18].

Rank-based differences also revealed important inequalities. In our investigation, the DMFT index (Table 3) and CI (Table 3) were found to be statistically significantly different between officers and the other two rank groups, with officers consistently demonstrating better oral health outcomes. In contrast, a Spanish Army personnel study found significant differences in troops compared with officers and non-commissioned officers (NCOs) [27]. Although troops comprise a rank group with a higher DMFT score in the Spanish Army [27], it is important to note that caries prevalence was nearly twice as great in Navy troops, with Army troops having a very low score and Navy troops having a low score of caries experience in relation to WHO criteria. This discrepancy highlights variations in oral health outcomes not only between ranks but also between different military branches. Additionally, disparities in rank may have a bearing on the process of establishing oral health as a priority [23], possibly linked to socioeconomic status, education level, and access to information and services.

Regarding place of origin, Navy personnel from Spain and other European countries exhibited significantly lower caries experience compared to those from outside Europe. The DMFT index showed statistically significant differences in out-of-Europe personnel. Similar findings were reported in a study of employed adults in Spain, in which individuals from other countries also had higher scores compared to those born in Spain [31]. Conversely, a study of Israel military recruits reported lower scores among native-born participants and higher scores among foreign-born individuals [32]. Additionally, in our study, significant differences according to unit were observed (Table 1). These findings underline the need to consider geographical and cultural backgrounds when designing preventive strategies.

This study presents findings of higher DMFT and CI scores, in comparison with data that has previously been published regarding military personnel [23,24,27,32]. From a clinical and operational viewpoint, elevated DMFT scores are associated with an increased risk of dental emergencies, which could compromise capability to complete missions. Consequently, it can be deduced that the deployment of a greater proportion of naval personnel with accumulated caries experience will result in an escalated risk of dental emergencies during military operations [2]. Oral emergencies, specifically caries and periodontal disease, are the main causes of dental care during Navy deployments [36,37,38]. For this reason, oral health prevention plans should be integrated into deployment preparation, with targeted interventions for high-risk groups (such as lower ranks, younger age groups, and non-European personnel), and oral health policy should reflect this stratification.

Several limitations of this study need to be acknowledged. First, there is a lack of prior oral health research on Spanish military personnel, specifically regarding caries experience in Navy personnel, making direct comparisons difficult. One significant limitation encountered during the course of the present study was the difficulty in obtaining a homogeneous sample with respect to gender, age, and rank. The majority of military personnel, especially within the Navy, are men, younger than 45 years old, and hold lower ranks, which may limit the generalizability of the findings to other sectors or countries [13,14]. Additionally, the retrospective design introduces potential biases. The retrospective design limits the ability to establish causality, as data were collected at a single point in time. Furthermore, this study used a census sampling approach, which, while covering the majority of dental records, excluded personnel who did not attend their mandatory pre-mission dental examination, potentially introducing selection and participation bias. Lastly, the cross-sectional nature of this study prevents the assessment of long-term trends or causal relationships. The findings may also be limited by the restricted comparison between the military population and the general civilian population. In future research, we will aim to conduct a randomized study with greater homogeneity between the samples of each group, although it is not easy to obtain large homogenous Navy personnel samples.

To the best of our knowledge, this is the first study to analyze caries experience in Spanish Navy personnel. Future research involving multiple centers, specialties, and larger samples could contribute to a more complete understanding of oral health course in diverse naval environments.

## 5. Conclusions

From the analysis carried out, it can be stated that this study found a low prevalence of cavitated caries compared to those reported globally, for employed adults in Spain, for Spanish Army personnel, and in another epidemiological study in Spain.

Although we did not find significant differences in caries prevalence according to gender, younger personnel and officers were observed to have lower DMFT scores and higher CIs. Furthermore, among men, non-commissioned officers showed the highest DMFT indexes.

From the results obtained, we reject the null hypothesis, which is that there are no significant differences in terms of caries experience and restorative care among the different groups of the population studied.

## Figures and Tables

**Table 1 jcm-14-05236-t001:** Comparison of DMFT values between groups according to each variable (*n* = 1318).

			DMFT			
		*n* (%)	Mean ± SD	95% CI	*p*-Value	Effect Size
Gender	Female	222 (16.8)	6.31 ± 4.34	5.74–6.89	0.267	
	Male	1096 (83.2)	5.92 ± 4.78	5.64–6.21		
Age	20–34 years	698 (83.2)	4.60 ± 4.12	4.30–4.90	<0.001 **	0.10
	35–44 years	398 (30.2)	7.13 ± 4.64	6.67–7.58		
	>45 years	222 (16.8)	8.33 ± 5.14	7.64–9.01		
Service	Navy on land	91 (6.9)	6.85 ± 4.83	5.84–7.85	0.068	
	Embarked Navy	1112 (84.4)	5.86 ± 4.67	5.59–6.13		
	Navy Air Force	115 (8.7)	6.54 ± 5.01	5.61–7.46		
Place of Origin	Spain	1219 (92.5)	5.96 ± 4.67	5.70–6.22	0.019 *	0.01
	Europe	85 (6.4)	5.85 ± 5.09	6.43–12.57		
	Outside Europe	14 (1.1)	9.50 ± 5.32	4.75–6.94		
Rank	Troop	897 (68.1)	6.00 ± 4.68	5.69–6.31	<0.001 **	0.01
	Non-commissioned officer	245 (18.6)	6.78 ± 4.82	6.18–7.39		
	Officer	176 (13.3)	4.83 ± 4.51	4.16–5.50		
Unit	BAA Castilla	189 (14.4)	6.34 ± 4.57	5.69–7.00	0.046 *	0.01
	Headquarter	114 (8.6)	6.86 ± 4.65	6.00–7.72		
	FG Navarra	234 (17.7)	5.82 ± 4.90	5.19–6.45		
	FG Victoria	207 (15.7)	5.96 ± 5.16	5.25–6.67		
	LHD Juan Carlos	321 (24.3)	5.47 ± 4.22	5.00–5.93		
	FG Canarias	79 (6.0)	5.15 ± 4.34	4.17–6.12		
	Aircraft squadrons	103 (7.8)	6.51 ± 5.10	5.17–7.51		
	Other ships	33 (2.5)	7.30 ± 4.56	5.69–8.92		
	Others	38 (2.9)	6.37 ± 5.20	4.66–8.08		

Statistically significant results (*: *p* < 0.05; **: *p* < 0.01); DMFT: Decayed, Missing, Filled Teeth; SD: standard deviation; CI: confidence interval.

**Table 2 jcm-14-05236-t002:** Comparison of DMFT values between groups according to rank and gender.

DMFT						
Gender	Rank	Mean	SD	95% CI	*p*-Value	Effect Size
Male	Troop	5.88	4.77	5.53–6.23	<0.001 **	0.01
	Non-Commissioned Officer	6.81	4.83	6.20–7.42		
	Officer	4.89	4.56	4.17–5.61		
Female	Troop	6.45	4.31	5.78–7.12	0.161	
	Non-Commissioned Officer	6.45	4.71	4.39–8.51		
	Officer	4.08	3.88	1.52–6.62		

** Statistically significant results (*p* < 0.01); DMFT: Decayed, Missing, Filled Teeth; SD: standard deviation; CI: confidence interval.

**Table 3 jcm-14-05236-t003:** Comparison of DMFT values among ranks depending on gender.

Gender	Rank	95% CI	*p*-Value
Male	Troop–NCO	−1.79–0.07	0.028 *
	Troop–Officer	0.02–1.97	0.045 *
	NCO–Officer	0.77–3.08	<0.001 **
Female	Troop–NCO	−2.64–2.64	1.000
	Troop–Officer	−0.85–5.60	0.233
	NCO–Officer	−5.59–0.89	0.466

Statistically significant results (*: *p* < 0.05; **: *p* < 0.01); NCO: non-commissioned officer; CI: confidence interval.

**Table 4 jcm-14-05236-t004:** Comparison of DMFT values between groups according to age and gender.

DMFT		Age (Years)				
Gender		20–34	35–44	>45	*p*-Value	Effect Size
Male	Mean	4.45	7.02	8.34	0.004 *	0.11
	SD	4.15	4.72	5.14		
Female	Mean	5.25	7.58	8.24	<0.001 **	0.08
	SD	3.92	4.26	5.31		

Statistically significant results (*: *p* < 0.05; **: *p* < 0.01); SD: standard deviation.

**Table 5 jcm-14-05236-t005:** Comparison of DMFT values among ranks depending on age groups.

DMFT		Rank				
Age (Years)		Troop	Non-Commissioned Officer	Officer	*p*-Value	Effect Size
20–34	Mean	4.86	4.89	2.56	<0.001 **	0.03
	SD	4.17	4.10	3.09		
35–44	Mean	7.42	7.08	5.48	0.041 *	0.02
	SD	4.55	5.05	3.86		
>45	Mean	8.59	8.36	7.79	0.660	
	SD	5.61	4.59	4.95		

Statistically significant results (*: *p* < 0.05; **: *p* < 0.01); SD: standard deviation.

**Table 6 jcm-14-05236-t006:** Comparison of CI values between groups of each variable (*n* = 1318).

			CI			
		*n* (%)	Mean ± SD	95% CI	*p*-Value	Effect Size
Gender	Female	222 (16.8)	79.94 ± 26.24	76.33–80.89	0.718	
	Male	1096 (83.2)	79.20 ± 26.60	77.51–80.89		
Age	20–34 years	698 (83.2)	83.70 ± 26.47	81.54–85.86	<0.001 **	
	35–44 years	398 (30.2)	75.69 ± 25.87	73.03–78.36		0.03
	>45 years	222 (16.8)	73.46 ± 25.89	69.94–76.99		
Service	Navy on land	91 (6.9)	81.43 ± 26.31	75.72–87.11	0.743	
	Embarked Navy	1112 (84.4)	79.26 ± 26.52	77.59–80.93		
	Navy Air Force	115 (8.7)	78.69 ± 26.81	73.48–83.91		
Place of Origin	Spain	1219 (92.5)	79.11 ± 26.71	77.51–80.71	0.478	
	Europe	85 (6.4)	83.04 ± 24.56	64.87–90.93		
	Outside Europe	14 (1.1)	77.90 ± 21.56	77.80–88.89		
Rank	Troop	897 (68.1)	78.21 ± 27.33	76.31–80.12	0.001 **	
	Non-commissioned officer	245 (18.6)	78.07 ± 23.69	74.90–81.24		0.01
	Officer	176 (13.3)	87.22 ± 24.91	83.16–81.24		
Unit	BAA Castilla	189 (14.4)	76.79 ± 27.02	72.64–80.49	0.926	
	Headquarter	114 (8.6)	79.32 ± 26.67	74.23–84.41		
	FG Navarra	234 (17.7)	79.21 ± 26.32	75.53–82.89		
	FG Victoria	207 (15.7)	78.99 ± 26.30	75.03–82.92		
	LHD Juan Carlos	321 (24.3)	80.38 ± 27.51	77.17–83.58		
	FG Canarias	79 (6.0)	82.03 ± 24.74	76.04–88.01		
	Aircraft squadrons	103 (7.8)	78.93 ± 25.06	73.74–84.12		
	Other ships	33 (2.5)	82.02 ± 19.73	74.91–89.14		
	Others	38 (2.9)	77.87 ± 32.19	66.46–89.29		

** Statistically significant results (*p* < 0.01); SD: standard deviation; CI: care index; 95% CI: confidence interval.

**Table 7 jcm-14-05236-t007:** Comparison of means between DMFT index and CI among groups of variables.

		DMFT		CI		*p*-Value
		Mean	SD	Mean	SD	
Rank	Troop–NCO	−0.762	0.337	0.140	2.021	0.722
	Troop–Officers	1.192	0.386	−9.011	2.369	0.001 **
	NCO–Officers	1.954	0.462	−9.151	2.818	0.010 **
Place of origin	Spain–Europe	−3.563	1.264	1.230	7.402	0.763
	Spain–Outside Europe	0.213	0.522	−4.141	3.251	0.179
	Europe–Outside Europe	3.776	1.354	−5.371	8.003	0.502
Age	20–34–35–44	−2.477	0.280	8.005	1.749	0.001 **
	20–34– >5	−3.731	0.345	10.235	2.108	0.001 **
	35–44– >45	−1.254	0.374	2.230	2.267	0.602

** Statistically significant results (*p* < 0.01); DMFT: Decayed, Missing, Filled Teeth; CI: care index; NCO: non-commissioned officer.

## Data Availability

The original contributions presented in this study are included in the article.

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
