# Peer review of "The Oral Health Status of Spanish Naval Military Personnel: A Retrospective Study"

_jcm, 2025, doi:10.3390/jcm14155236_

Round 1

Reviewer 1 Report

Comments and Suggestions for Authors

Dear authors, I appreciate your submission. This research investigates a significant and often overlooked subject—oral health status among personnel in the Spanish Navy—and utilizes a relatively extensive and valuable dataset. Nevertheless, various elements of the manuscript necessitate revision and clarification. Please take into account the following observations:

1) The introduction offers pertinent background information but could benefit from enhanced structure and clarity. Key points are occasionally repetitive or loosely connected. I suggest clearly delineating: the existing gap in the literature, the specific significance of oral health within naval military contexts, and the justification for selecting DMFT and CI indices.

2) The study design is generally suitable; however, the explanation of sample selection, inclusion/exclusion criteria, and dental examination protocol requires greater precision: Who conducted the dental examinations? Were they standardized? Clarify whether intra- or inter-examiner reliability was evaluated. Provide additional details regarding the randomization process employed for sampling.

3) The tables are informative, yet several are excessively dense and challenging to interpret. Consider summarizing key findings within the text and relocating detailed comparisons to supplementary materials. The statistical methods are generally appropriate, but p-values alone are inadequate. Please incorporate effect sizes and confidence intervals where applicable. Reorganize the results to enhance flow (e.g., present all DMFT results initially, followed by CI). Ensure table formatting is consistent (e.g., decimal points, units, alignment).

4) The discussion includes numerous valuable comparative data points but lacks critical depth. Elucidate the clinical significance of your findings (e.g., what does a CI of 79% indicate for readiness or policy?). Refrain from excessive listing of international studies unless they are directly related to your findings. Provide a more comprehensive interpretation of rank-based differences and potential socioeconomic influences.

5) The authors succinctly acknowledge the limitations of the study, specifically regarding sample homogeneity and gender imbalance. I suggest that this section be elaborated to encompass: Possible biases arising from the retrospective design, constraints in generalizing findings to other sectors or nations, and the absence of longitudinal data.

Kind regards!

Comments on the Quality of English Language

The manuscript requires an extensive revision of its language. It is rife with grammatical errors, awkward phrasing, and inconsistencies (for example, "scubdivers" instead of "scuba divers"; "valability dental services"). I suggest employing a professional English language editing service prior to resubmission.

Author Response

Dear Reviewer,

Thanks for taking the time to review our manuscript and suggesting improvements to our work by providing such detail. We have revised the manuscript and addressed the points you and others have raised. We are submitting what we think is a much-improved version of our work. This version includes improvements in clarity, quality, English grammar, and structure. Below, we provide point-by-point responses to each of your suggestions, outlining the corresponding revisions made to the manuscript.

Sincerely,

Bárbara Manso de Gustín

Dear authors, I appreciate your submission. This research investigates a significant and often overlooked subject—oral health status among personnel in the Spanish Navy—and utilizes a relatively extensive and valuable dataset. Nevertheless, various elements of the manuscript necessitate revision and clarification. Please take into account the following observations:

1. The introduction offers pertinent background information but could benefit fromNenhanced structure and clarity. Key points are occasionally repetitive or loosely connected. I suggest clearly delineating: the existing gap in the literature, the specific significance of oral health within naval military contexts, and the justification for selecting DMFT and CI indices.

The reviewer’s observation is appreciated. The authors have restructured the introduction to clearly highlight the gap in the literature on naval personnel (Lines 50-52; 100), the importance of oral health in naval contexts (Lines 92-99), and why DMFT and CI are chosen (Lines 68-75).

2. The study design is generally suitable; however, the explanation of sample selection, inclusion/exclusion criteria, and dental examination protocol requires greater precision: Who conducted the dental examinations? Were they standardized? Clarify whether intra- or inter-examiner reliability was evaluated. Provide additional details regarding the randomization process employed for sampling.

The authors welcome the reviewer’s input and have addressed it accordingly. The Methods section has been revised to include new paragraphs addressing this comment. The changes made are in Sections 2.1 and 2.2.

3. The tables are informative, yet several are excessively dense and challenging to interpret. Consider summarizing key findings within the text and relocating detailed comparisons to supplementary materials. The statistical methods are generally appropriate, but p-values alone are inadequate. Please incorporate effect sizes and confidence intervals where applicable. Reorganize the results to enhance flow (e.g.,present all DMFT results initially, followed by CI). Ensure table formatting is consistent (e.g., decimal points, units, alignment).

The authors thank the reviewer for the valuable and insightful comments. In response, several tables have been revised and simplified to enhance clarity and readability. Effect sizes and 95% confidence intervals have been included for key comparisons. Furthermore, to improve the statistical reporting, more precise p-values have been incorporated into the revised Tables 2, 4 and 5. In accordance with Reviewer 2’s recommendation, the previous Tables 3, 4, and 5 have been merged into a single, more concise table (now Table 7). Table 10 has been removed, as it was not considered relevant to the main findings. The Results section has been reorganized to present DMFT results first, followed by CI results, ensuring a more logical flow. Finally, table formatting has been reviewed and standardized throughout.

4. The discussion includes numerous valuable comparative data points but lacks critical depth. Elucidate the clinical significance of your findings (e.g., what does a CI of 79% indicate for readiness or policy?). Refrain from excessive listing of international studies unless they are directly related to your findings. Provide a more comprehensive interpretation of rank-based differences and potential socioeconomic influences.

The Discussion now includes practical implications of the CI score, fewer unrelated comparisons, and a more in-depth explanation of rank-based and socioeconomic factors. The changes made are in Section 4.

5. The authors succinctly acknowledge the limitations of the study, specifically regarding sample homogeneity and gender imbalance. I suggest that this section be elaborated to encompass: Possible biases arising from the retrospective design, constraints in generalizing findings to other sectors or nations, and the absence of longitudinal data.

Thank you for your careful evaluation and recommendations. The authors have expanded the Limitations section to address retrospective design bias, generalizability limits, and absence of follow-up data (Lines 332-345).

The manuscript requires an extensive revision of its language. It is rife with grammatical errors, awkward phrasing, and inconsistencies (for example, "scubdivers" instead of "scuba divers"; "valability dental services"). I suggest employing a professional English language editing service prior to resubmission.

We sincerely thank reviewer for the observation regarding the language quality of the manuscript. The authors confirm that the manuscript has undergone comprehensive professional editing. Specifically, it was revised through MDPI Author Services for Layout Editing on June 10, 2025, and subsequently through English Language Editing by MDPI on July 16, 2025. Grammatical and syntactic errors and awkward phrasing have been corrected throughout the manuscript. The corresponding editing confirmation documents are attached for reference.

English Layout EditingEnglish Language Editing

Reviewer 2 Report

Comments and Suggestions for Authors

This manuscript explores a relevant yet insufficiently studied area—oral health status among Spanish naval military personnel. It outlines a well-defined objective with an appropriate retrospective design, and is based on a relatively large and clearly characterized sample.

My comments are below.

  1. The manuscript contains numerous grammatical and syntactic errors. Comprehensive English language editing or proofreading by a native speaker or professional editor is strongly recommended to ensure clarity and readability throughout.
  2. In the introduction, the sentence Oral diseases, such as tooth decay and tooth loss, serve as clinical indicators of suboptimal oral health” could be misleading. Tooth loss itself is not an oral disease but rather a consequence, most commonly of periodontitis.
    Even though periodontitis is not the main focus of this study, it would be appropriate to briefly mention it as one of the most frequent causes of tooth loss in adults.
  3. In the sentence: “Regarding to Spain the average number of (Decayed, missing, filled teeth) DMFT index…”, the use of parentheses is unnecessary and grammatically incorrect.

  4. In sentence: “Caries experience is typically measured using the DMFT index, which calculates the total number of teeth with decayed (D), missing (M) due to caries and filled (F) teeth (DMFT)”, the acronym DMFT at the end is redundant and should be removed.
  5. The sentence:“For this study, a total of 1318 average range was between 21 and 56 and dental records were used” is grammatically incorrect and unclear.
  6. The following sentence is repetitive and unclear:
    “The CI was calculated by determining the number of restored teeth was calculated by determining the number of restored teeth as a proportion…”

    May be rewritten as: “The CI was calculated by determining the number of restored teeth as a proportion of the total number of decayed (D), missing (M), and filled (F) teeth, using the formula: CI = F / DMFT × 100%.”

  7. The data presented in Tables 3, 4, and 5 could be more effectively presented as a single table.
  8. Did the available dental records include information on oral hygiene habits or smoking status? Including such lifestyle factors could substantially strengthen the interpretation of DMFT and CI outcomes, given their known influence on oral health.
  9. Presenting the limitations in a single, clearly defined paragraph would improve the structure of the discussion section, as they are currently mentioned in a fragmented manner.
Comments on the Quality of English Language

The manuscript contains numerous grammatical and syntactic errors. Comprehensive English language editing or proofreading by a native speaker or professional editor is strongly recommended to ensure clarity and readability throughout.

Author Response

Dear Reviewer,

Thanks for taking the time to review our manuscript and suggesting improvements to our work by providing such detail. We have revised the manuscript and addressed the points you and others have raised. We are submitting what we think is a much-improved version of our work. This version includes improvements in clarity, quality, English grammar, and structure. Below, we provide point-by-point responses to each of your suggestions, outlining the corresponding revisions made to the manuscript.

Sincerely,

Bárbara Manso de Gustín

This manuscript explores a relevant yet insufficiently studied area—oral health status among Spanish naval military personnel. It outlines a well-defined objective with an appropriate retrospective design and is based on a relatively large and clearly characterized sample.

My comments are below.

1. The manuscript contains numerous grammatical and syntactic errors. Comprehensive English language editing or proofreading by a native speaker or professional editor is strongly recommended to ensure clarity and readability throughout.

We sincerely thank reviewer for the observation regarding the language quality of the manuscript. The authors confirm that the manuscript has undergone comprehensive professional editing. Specifically, it was revised through MDPI Author Services for Layout Editing on June 10, 2025, and subsequently through English Language Editing by MDPI on July 16, 2025. Grammatical and syntactic errors and awkward phrasing have been corrected throughout the manuscript. The corresponding editing confirmation documents are attached for reference.

2. In the introduction, the sentence “Oral diseases, such as tooth decay and tooth loss, serve as clinical indicators of suboptimal oral health” could be misleading. Tooth loss itself is not an oral disease but rather a consequence, most commonly of periodontitis.

Even though periodontitis is not the main focus of this study, it would be appropriate to briefly mention it as one of the most frequent causes of tooth loss in adults.

The reviewer’s insightful comment is acknowledged. The authors have revised the sentence and reflected periodontitis as a leading cause (Lines 40-42).

3. In the sentence: “Regarding to Spain the average number of (Decayed, missing, filled teeth) DMFT index…”, the use of parentheses is unnecessary and grammatically incorrect.

Done (Lines 58-60).

4. In sentence: “Caries experience is typically measured using the DMFT index, which calculates the total number of teeth with decayed (D), missing (M) due to caries and filled (F) teeth (DMFT)”, the acronym DMFT at the end is redundant and should be removed.

The reviewer’s observation is appreciated. The acronym DMFT has been removed (Lines 65-67).

5. The sentence: “For this study, a total of 1318 average range was between 21 and 56 and dental records were used” is grammatically incorrect and unclear.

The sentence has been rewritten for clarity and correctness (Lines 115-117).

6.The following sentence is repetitive and unclear: “The CI was calculated by determining the number of restored teeth was calculated by determining the number of restored teeth as a proportion…”

May be rewritten as: “The CI was calculated by determining the number of restored teeth as a proportion of the total number of decayed (D), missing (M), and filled (F) teeth, using the formula: CI = F / DMFT × 100%.”

Done (Lines 156-158).

7. The data presented in Tables 3, 4, and 5 could be more effectively presented as a single table.

The suggestions provided are highly appreciated. The authors have merged the contents of Tables 3, 4, and 5 into a single, consolidated table (now Table 7) to enhance clarity and comparison (Section 3, Table 7).

8. Did the available dental records include information on oral hygiene habits or smoking status? Including such lifestyle factors could substantially strengthen the interpretation of DMFT and CI outcomes, given their known influence on oral health.

Thank you for this suggestion. Unfortunately, the available records did not include data on oral hygiene habits or smoking status. The authors acknowledge their relevance, and future studies should consider these variables to better contextualize oral health outcomes in this population.

9. Presenting the limitations in a single, clearly defined paragraph would improve the structure of the discussion section, as they are currently mentioned in a fragmented manner.

Done (Lines 332-349).

English Layout Editing

English Language Editing

Round 2

Reviewer 1 Report

Comments and Suggestions for Authors

Dear authors, I have no further objections. Congrats!